# General Practitioners’ Attitudes toward Municipal Initiatives to Improve Antibiotic Prescribing—A Mixed-Methods Study

**DOI:** 10.3390/antibiotics8030120

**Published:** 2019-08-17

**Authors:** Marthe Sunde, Marthe Marie Nygaard, Sigurd Høye

**Affiliations:** 1Faculty of Medicine, University of Oslo, 0318 Oslo, Norway; 2Antibiotic Centre for Primary Care, Department of General Practice, Institute of Health and Society, University of Oslo, 0318 Oslo, Norway

**Keywords:** quality improvement, general practitioners, primary care, antibiotics, guideline

## Abstract

Antimicrobial stewardship (AMS) interventions directed at general practitioners (GPs) contribute to an improved antibiotic prescribing. However, it is challenging to implement and maintain such interventions at a national level. Involving the municipalities’ Chief Medical Officer (MCMO) in quality improvement activities may simplify the implementation and maintenance, but may also be perceived challenging for the GPs. In the ENORM (Educational intervention in NORwegian Municipalities for antibiotic treatment in line with guidelines) study, MCMOs acted as facilitators of an AMS intervention for GPs. We explored GPs’ views on their own antibiotic prescribing, and their views on MCMO involvement in improving antibiotic prescribing in general practice. This is a mixed-methods study combining quantitative and qualitative data from two data sources: e-mail interviews with 15 GPs prior to the ENORM intervention, and online-form answers to closed and open-ended questions from 132 GPs participating in the ENORM intervention. The interviews and open-ended responses were analyzed using systematic text condensation. Many GPs admitted to occasionally prescribing antibiotics without medical indication, mainly due to pressure from patients. Too liberal treatment guidelines were also seen as a reason for overtreatment. The MCMO was considered a suitable and acceptable facilitator of quality improvement activities in general practice, and their involvement was regarded as unproblematic (scale 0 (very problematic) to 10 (not problematic at all): mean 8.2, median 10). GPs acknowledge the need and possibility to improve their own antibiotic prescribing, and in doing so, they welcome engagement from the municipality. MCMOs should be involved in quality improvement and AMS in general practice.

## 1. Introduction

The presence of antibiotic resistance is closely connected to the consumption of antibiotics, both at a societal and an individual level, and most antibiotics consumed by humans are prescribed in primary care [1]. Compared to other European countries, the use of antibiotics in Norwegian primary care is relatively low, but there is still room for improvement [2]. Accordingly, in 2016 the Norwegian Ministry of Health and Care Services published an action plan with the aim of an overall 30% reduction in the use of antibiotics by 2020 [3].

As general practitioners (GPs) have a key role in regulating the amount of consumed antibiotics in primary care, numerous antimicrobial stewardship interventions directed at GPs have been tried out. In the Norwegian *Rx*-PAD (Prescription Peer Academic Detailing) study [4], effective strategies such as educational outreach visits [5] and audit and feedback [6] were combined, resulting in a relative reduction in antibiotic prescribing of 9%, and an increased proportion of narrow spectrum antibiotic of 24%.

The participating GPs found the intervention acceptable [7], and especially valued that the intervention was implemented by a peer GP and not by an external authority.

In Norway, GPs are mainly self-employed and have no clinical superiors. Professional autonomy is highly valued [8]. The municipality, with which the GP has a contract to deliver primary care services, has a legal responsibility to ensure quality improvement and patient safety measures for the GPs [9]. However, how this responsibility should be interpreted and implemented is very unclear [10].

The *Rx*-PAD intervention was resource intensive, as it included training, travel expenses, and payment for the GPs who acted as peer academic detailers. Hence, the intervention has not been implemented as a freely available course for Norwegian GPs. The same is the case for many antimicrobial stewardship interventions both in primary and secondary care. To overcome these challenges, we developed the ENORM (Educational intervention in NORwegian Municipalities for antibiotic treatment in line with guidelines) study [11], which aims at testing an antimicrobial stewardship intervention that, if effective and acceptable, can be easily implemented throughout primary care. The intervention; a 15-hour course consisting of 3 e-learning sessions and 3 Continuing Medical Education (CME) group meetings where the GPs are to discuss their personal prescription reports, is rolled out by a representative of the municipality, the Municipal Chief Medical Officer (MCMO). In small municipalities, the MCMO post usually is a part-time position for a GP, while full-time MCMO position is the norm in large municipalities. The ENORM intervention was tested in 30 randomly selected, medium-size (5.000–25.000 inhabitants), high-antibiotic-consumption Norwegian municipalities between February 2017 and September 2018. The effect of the intervention will be reported in a separate publication.

The aim of this study is to explore the GPs’ judgment of their own antibiotic prescribing, and their views and attitudes toward MCMOs’ involvement in antimicrobial stewardship efforts toward GPs.

## 2. Results

### 2.1. Part 1—E-Mail Interviews

During our analysis, we defined three main themes: antibiotic prescribing, reduction, and quality improvement.

#### 2.1.1. Antibiotic Prescribing

The GPs were divided in their views of own antibiotic prescribing without medical indication. Many stated that this was rarely or never the case. An equally common statement was that the GP admitted to doing this on some occasions. A view was held that the correct prescription practice is not easily defined:

“I don’t prescribe antibiotics without a medical indication, but here I mean there are grey areas of right/wrong use, and not a black/white picture on indication.” (GP, board member)

Perceived pressure from patients was the most common explanation as to why the GPs prescribed antibiotics without medical indication. While some GPs had recently experienced patients to be more knowledgeable in their views on antibiotic use, the majority still experienced pressure from patients to prescribe antibiotics. They expressed that this demand could be challenging to withstand. Fear of under-treating, the patient’s strong opinion of the need for antibiotic treatment, and lack of time to explain and convince the patient that the treatment is unnecessary, were pointed out as reasons for not withstanding the pressure. Some pointed out patients’ lack of understanding the difference between bacterial and viral infections, and a perception of antibiotics as an easy quick-fix.

Treatment guidelines were mentioned within the topic of reasons for antibiotic overuse. Some GPs claimed that the actual guidelines on antibiotic use are an obstacle for reducing antibiotic prescribing, and pointed to several areas where they found that this was the case. Otitis, respiratory tract infections, uncomplicated cystitis, and acne were some of the indications suggested for revision.

”I think (...) the indications are too spacious. We are over-treating when we follow guidelines.” (GP with board membership)

#### 2.1.2. Reduction

The GPs held that antimicrobial resistance is a serious threat, and they strongly agreed that antibiotic use had to be greatly reduced. The GPs that reflected on their own ability to contribute to reducing prescription rates were concerned about the goal being set too high. The majority stated that they were unsure of their ability to cut their own prescription rates to match the 30% reduction goal set by the Government. One GP stated that their prescription rate was at a level where 30% reduction would not be a justifiable correction.

The GPs had several ideas on how to reduce antibiotic prescribing. Guideline revision, information to the patients and public on antibiotic resistance, and the use of delayed prescribing were suggested. Although most examples of ways to reduce antibiotic prescriptions were very specific, one GP had thoughts on a more long-term measure:

“Make the GPs proud of their professional mandate, including their ability to single out the few seriously ill patients who actually need antibiotics.” (GP with board membership)

This view was opposed by another GP who pointed out how their professional freedom is rightfully challenged when matters of serious public concern are debated: 

“As all my colleagues, I value my independence as a general practitioner, but certain cases (like the problem of bacterial resistance) are too important not to manage centrally.” (GP without board membership)

#### 2.1.3. Quality Improvement

The GPs welcomed the MCMO as a facilitator of quality improvement activities in primary care. It was described as a praiseworthy and reasonable use of the MCMO’s time, and an interesting quality improvement approach. Some informants explicitly missed the municipalities’ engagement in quality improvement.

“(If the municipality initiated quality improvement measures) I would probably faint by the surprise of it. My municipality does nothing to improve the content of the primary care service.” (GP with board membership)

A view was held that the MCMO would be in an appropriate professional position to initiate quality improvement initiatives toward GPs.

“Professional independence is a sensitive topic for general practitioners. An arrangement like this by the pharmacy, for an example, would end up wrong. A municipality doctor or infection doctor would possibly be acceptable.” (GP without board membership)

### 2.2. Part 2—Online-Form Answers

#### 2.2.1. Own Potential for Improvement

We received 132 answers to the question “What do you consider to be your potential for improvement?” All respondents presented one or several specific goals for their antibiotic prescribing. They pointed out specific diagnoses and age groups where they could reduce their use of antibiotics, and considered tools to help them achieving their goals, such as enhanced consultation techniques. Some of the respondents also set a time limit for when they should have reached their goals. Even though the specificity and comprehensiveness of the answers varied, no respondents argued that they saw small, or no, room for improvement.

#### 2.2.2. The Municipal Chief Medical Officer’s Role

About 85% (112) of the 132 participants responded to the rating scale question “What is your opinion on the Municipal Chief Medical Officer’s role in this course?” On the 11 points scale (0: very problematic to 10: not problematic at all), the mean value was 8.2 (95% CI: 7.6–8.7), and the median value was 10.

Twenty five percent (33) of the 132 participants responded to the open-ended question “Provide comments on the Municipal Chief Medical Officer’s role.” Four respondents commented that their MCMO did not have any role in the implementation of the ENORM course in their CME group, and five respondents commented that they themselves were MCMOs, in addition to being GPs. The remaining 24 respondents all gave positive comments on the MCMOs role. Two main explanations were identified: First, the participants appreciated having a designated course facilitator. They experienced that the MCMO coordinated and structured the meeting, gave useful information and reminders, and initiated valuable group discussions with relevant and relatable cases. Second, it was appreciated that the MCMO engaged in the issue of antibiotic prescribing and set an example by recommending national guidelines.

Some of the respondents commented that the MCMO also participated in the course as a GP, or that they knew them as a GP, and that they considered the MCMO to be more a peer than a manager.

## 3. Discussion

### 3.1. Main Findings

The general practitioners in this study stated that there is an overuse of antibiotics in general practice, and that the use should be reduced. Perceived pressure from patients, lack of time, fear of under-treatment, and too liberal guidelines were pointed out as drivers of unnecessary antibiotic prescribing. Both arguments that GPs themselves could bring about the needed reduction and that external efforts were justified for GPs to reduce prescribing existed among the GPs. The Municipal Chief Medical Officer (MCMO) was seen as a suitable and acceptable facilitator of quality improvement activities in general practice. This view was supported by GPs taking part in the ENORM intervention, facilitated by their MCMO.

### 3.2. Strengths and Limitations

This study has some limitations. The response rate on the open-ended questions was relatively low in both part 1 (15% of invited GPs without board position) and part 2 (25% of participating GPs) of the study. GPs with strong views on antimicrobial stewardship and GP’s autonomy, work frames, and quality improvement may have been more inclined to participate, as suggested by the fact that 62% of GPs with board positions responded. However, the aim in qualitative studies is not to recruit a representative sample of informants within the given population, but to explore existing views among a variety of informants [12]. We find that the selection of GPs was diverse, including both opinion leaders and GPs without any board membership, and we achieved variety with respect to geography, gender, municipality’s size, and municipality’s antibiotic consumption.

Our selection of informants is quite small. In a paper on sample sizes in qualitative studies [13], Malterud presents “information power” as a concept. Malterud suggests that the more relevant information a sample holds for a study, the lower number of participants is needed. Our informants may be considered to provide information of high value, as they included both potential participants of a planned intervention and participants of the actual intervention.

The respondents gave relatively short answers to the open-ended questions, ranging from 1 to 180 words. As held by Meho [14], e-mail interviewing can be a viable alternative to face-to-face interviews in qualitative research. The opportunity of easy outreach to informants over a large geographic area was a great benefit in our study. The challenge of using this method, however, is the less opportunity for continuous correspondence, and the information gathered is less in-depth than what could be expected in a face-to-face interview. We also experienced that some GPs agreed to participate but did not respond when the e-mail with the actual interview was sent, suggesting that e-mail interviews may be perceived as less binding than a face-to-face interview.

Due to the wording of the question “What do you consider to be your potential for improvement?” respondents may have felt obliged to suggest such a potential, even if they found that there was no need for improvement. Also, the respondents in part 2 had agreed to participate in a course facilitated by the MCMO. GPs with negative views on such cooperation may have rejected the MCMOs invitation, thus introducing a selection bias in this part of the study. However, no negative remarks regarding the MCMOs’ involvement were expressed in part 1 of the study.

### 3.3. Results and Discussion

Many of the informants admitted to occasionally prescribing antibiotics without clinical indication. Their reasons for doing so were to a large extent coherent with previous findings on “non-pharmacological” antibiotic prescribing [15]: the lack of opportunity for continued care, pressure from patients, and the GP’s uncertainty. However, the view that antibiotic overtreatment may be due to adherence to guidelines is to our knowledge a new finding. GPs experience that obligations to adhere to several single-disease guidelines may lead to polypharmacy and overtreatment in multimorbid patients [16]. Especially, guidelines on prevention of cardiovascular disease are held to be inadequate and cause overtreatment in primary care [17]. The existence of this view among GPs when it comes to guidelines for antibiotic use may indicate that the guidelines could be stricter, but at the same time it also indicates that there is a risk of under-treatment of infections in primary care.

The MCMO’s role as facilitator and organizer of quality improvement activities in general practice was approved, both as a general idea among GPs not exposed to such activities, and based on the experience of GPs taking part in the ENORM intervention. Previous research has demonstrated the importance for GPs that quality improvement tutors are “one of them,” and independent of health authorities [7]. The MCMO is indeed a representative of the health authorities, but to a large extent, the participants regarded the MCMO as a peer, since many of them worked as both GP and MCMO. Also, the MCMOs in the ENORM intervention seem to have acted as facilitators, organizers, and initiators rather than experts or superiors.

There is an ongoing frustration among GPs, both in Norway and internationally, that GPs are imposed an increasing number of tasks and obligations without corresponding financial and organizational support [18,19]. The participants in our study valued the engagement from the municipality, possibly indicating that quality improvement and antimicrobial stewardship are perceived as necessary and wanted activities within general practice and not as obligations imposed by external authorities.

Preliminary results from part 1 of this study have been utilized to develop and tailor the content and implementation of the ENORM intervention.

## 4. Materials and Methods

This is a mixed-methods study combining quantitative and qualitative data from two data sources:

Part 1: E-mail interviews with a sample of GPs prior to the ENORM intervention.

Part 2: Online-form answers to closed and open-ended questions from GPs participating in the ENORM intervention.

### 4.1. Part 1—E-Mail Interviews

#### 4.1.1. Interview Guide

We developed a list of open-ended questions covering the themes—unnecessary antibiotic prescribing, own potential for reducing antibiotic prescribing, and the MCMO’s potential role in quality improvement in primary care. The questions were piloted among three GPs and modified according to the GPs’ comments (Appendix A).

#### 4.1.2. Selection and Recruitment

Two groups of informants were recruited:

(1) GPs who were considered to be opinion leaders among their peers, especially concerning clinical practice and quality improvement; board members of the General Practitioners’ Association (Allmennlegeforeningen), the Norwegian College of General Practitioners (Norsk forening for allmennmedisin) and the General Practice committee for Quality and Patient Safety (Allmennmedisinsk utvalg for kvalitet og pasientsikkerhet), a subcommittee of the Norwegian College of General Practitioners.

(2) GPs in high-antibiotic-consuming, medium-size (5.000–25.000 inhabitants) municipalities, aiming at variety with respect to geography, gender, municipality size, and municipality antibiotic consumption.

Invitations were sent out by either mail or e-mail. GPs that consented to participate were sent the questions by e-mail, and asked to reply to the e-mail. We aimed at about equal numbers of informants in group (1) and (2), and continued to recruit GPs to group (2) until this was achieved. In group (1), 8 out of 13 GPs consented to participate, and in group (2), 7 out of 47 GPs consented to participate, resulting in a total of 15 respondents. The interviews were performed between June 2016 and June 2017. None of the informants were exposed to the ENORM intervention.

#### 4.1.3. Analysis

The e-mail answers were anonymized and then analyzed using systematic text condensation [20]. MS and SH read all the text thoroughly, and agreed upon a list of initial themes, resulting in a coding frame. MS coded the text. The analysis followed four steps: (1) reading the complete material to obtain an overall impression; (2) identifying units of meaning representing different aspects of antibiotic use and antimicrobial stewardship efforts, and coding for these units; (3) condensing and summarizing the contents of each of the coded groups; and (4) generalizing descriptions and concepts. MS and SH read the condensates and agreed to the analysis. Illustrative quotes were translated into English. The software tool NVIVO was used in the analysis.

### 4.2. Part 2—Online-Form Answers

#### 4.2.1. Online Form

We developed an online form to be filled out by the participating GPs in the ENORM intervention directly after the first group meeting. The form had three purposes—to register completion of each participant’s group meeting, to aid as a quality improvement tool as part of a Plan-Do-Study-Act (PDSA) cycle [21], and to register the participants views on the Municipal Chief Medical Officers’ role in the intervention. Three of the online-form questions are used in this study:What do you consider to be your potential for improvement?What is your opinion on the Municipal Chief Medical Officer’s role in this course? (11-points rating scale, from 0 (very problematic) to 10 (not problematic at all))Provide comments on the Municipal Chief Medical Officer’s role.

The first question was mandatory in order to register completion of the group meeting, while the two other questions were optional. Hence, all GPs participating in the ENORM intervention were asked to fill the online form.

#### 4.2.2. Selection, Recruitment, and Analysis

MCMOs in the 30 ENORM intervention municipalities were informed that they belonged to a high-antibiotic-consuming municipality, and invited to recruit their local GPs to participate in a 15-hour antibiotic-prescribing improvement course. MCMOs who were also GPs could participate alongside the other GPs in their municipality. Participating GPs gave written consent to analysis of their online-form answers. By the end of the ENORM intervention in September 2018, 132 GPs had completed the first group meeting, resulting in 132 online-form answers.

The material was analyzed in the same manner as described in Part 1. MMN and SH read through all the responses, and MMN coded the text. The rating scale answers were analyzed using Microsoft Excel 2016.

### 4.3. Ethics (Part 1 and 2)

All participants gave informed consent to the study. The ENORM project was presented for the Regional Ethics Committee South/East (2016/1491/REK sør-øst C), which concluded that ethical approval was not mandatory. Data protection was approved by NSD—Norwegian Centre for Research Data (48136/3 and 50740/3).

## 5. Conclusions

GPs acknowledge the need and possibility to improve their own antibiotic prescribing. In doing so, they welcome engagement from the municipality, and they find that the MCMO is a suitable and acceptable quality improvement facilitator. MCMOs should be involved in quality improvement and antimicrobial stewardship efforts in general practice.

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
