# Peer review of "General Practitioners’ Attitudes toward Municipal Initiatives to Improve Antibiotic Prescribing—A Mixed-Methods Study"

_antibiotics, 2019, doi:10.3390/antibiotics8030120_

Round 1
Reviewer 1 Report
Peer review of: (31. July 2019).
“General practitioners’ attitudes towards external efforts to improve antibiotic prescribing – a mixed methods study” for Antibiotics, MDPI
I have sympathy with the purpose of the paper – find new ways to improve antibiotic prescribing in general practice.
I have some problems with the study:
The low response rate of 15 and 25% in the two parts of the study. And no really drop-out analysis of responders and non-responders. That means to me it is unclear how representative the study populations are. Could we have a table on that? The included Municipal Chief Medical Officers are both an object for some questions and responders. What does that mean? It is confusing. Do the high antibiotic consuming GPs know that they are invited because of high consuming? The mentioned ENORM intervention study is not described in detail. Reference 4 and 7 are the same.My conclusion revision is needed.
Author Response
Resubmission of manuscript General practitioners’ attitudes towards municipal initiatives to improve antibiotic prescribing – a mixed methods study, Antibiotics-560556
Dear editor and reviewers
Thank you for the opportunity to revise our manuscript. We appreciate the constructive suggestions. It is our belief that the manuscript is substantially improved after making the suggested edits.
Following this letter are the reviewer comments with our responses, including how and where the text was modified. Changes made in the manuscript are marked using track changes. Each author has given approval to the final form of this revision.
Reviewer 1
The low response rate of 15 and 25% in the two parts of the study. And no really drop-out analysis of responders and non-responders. That means to me it is unclear how representative the study populations are. Could we have a table on that?In this study, we chose to report the response rates, mainly because we combined qualitative and quantitative data. In my experience, the reponse rate is often not reported in papers based on qualitative studies, based on the fact that informants are selected strategically (with the aim of variety) rather than randomly. Accordingly, with a non-random selection, representativity is not achievable. The aim is rather to explore what kind of different views and experiences that exist among a variety of GPs.
We have explained this in more detail in the manuscript (line 175-179), and we have added a reference. We suggest not to include a table on the characteristics of responders and non-responders, be we are of course willing to add such a table if the editor and/or reviewer find it necessary.
The included Municipal Chief Medical Officers are both an object for some questions and responders. What does that mean? It is confusing.
We agree that this is confusing. Some of the MCMOs were also GPs, and participated in the intervention alongside the other GPs in their Continuing Medical Education group. We have tried to explain this in the manuscript (Line 149 and line 283)
Do the high antibiotic consuming GPs know that they are invited because of high consuming?
In the invitation to the ENORM intervention, the MCMOs and GPs were told that they were invited because their municpality was a high comsuming municipality (in average 15% higher consumption than the mean). We have added some information in the manuscript (Line 281)
The mentioned ENORM intervention study is not described in detail.
We are not sure how detailed we should describe the intervention. We have added some information in the manuscript (Line 61-68) and added a reference.
Reference 4 and 7 are the same.
Thank you for making us aware of this. It is fixed in the manuscript.
Reviewer 2
This is an interesting study that aims to explore the GP´s judgement of their own antibiotics prescribing, and their own views and attitudes towerds MCMOs´onvolvement in antimicrobial stewardships efforts towerds GPS. Attending to this objective and the methods developed, the title of the manuscript should be changed. "External efforts" implies that a variety of external efforts were explored and it is not true.We suggest the title “General practitioners’ attitudes towards municipal initiatives to improve antibiotic prescribing – a mixed methods study”
Lines 247-251 - Why groups of 47 informants were recruited for Group 1) and 13 for Group 2)? Did this GPS participated on ENORM intervention?
We aimed at about equal numbers of informants in group 1) and 2). A large proportion of eligible GPs in group 1) consented to participate, so we had to continue inviting GPs to group 2) until equal numbers were achieved. None of the informants participated in the ENORM intervention. We have added information on this in the manuscript (Line 251-255).
Was the Online form sended to all the particpants of the ENORM intervention?
Yes. Information on this is added in the manuscript (Line 277 and 286)
The last paragraph in the methods Ethics section is not necessary (lines 286-291). Please remove it
Thank you for making us aware of this. The section is removed.
Please add the Interview Guide as appedndix.
The interview guide is translated into English and added as appendix; Line 238: “(Appendix 1)”
Sincerely,
Sigurd Høye,
on behalf of the authors.
Reviewer 2 Report
This is an interesting study that aims to explore the GP´s judgement of their own antibiotics prescribing, and their own views and attitudes towerds MCMOs´onvolvement in antimicrobial stewardships efforts towerds GPS. Attending to this objective and the methods developed, the title of the manuscript should be changed. "External efforts" implies that a variety of external efforts were explored and it is not true.
Lines 247-251 - Why groups of 47 informants were recruited for Group 1) and 13 for Group 2)? Did this GPS participated on ENORM intervention?
Was the Online form sended to all the particpants of the ENORM intervention?
The last paragraph in the methods Ethics section is not necessary (lines 286-291). Please remove it
Please add the Interview Guide as appedndix.
Author Response

(The authors gave the same response as above.)

Round 2
Reviewer 1 Report
After revision.
I have now read the authors response and the new version of the paper. And I am satisfied with both.
I will recommend publication